# Description of a new fossil Thelyphonida (Arachnida, Uropygi) and further record of *Cratosolpuga wunderlichi* Selden, in Selden and Shear, 1996 (Arachnida, Solifugae) from Crato Formation (Aptian/Albian), Araripe Basin, Brazil

William Santana*, Allysson P. Pinheiro, Thiago Andrade Silva and Daniel Lima*

Museu de Paleontologia Plácido Cidade Nuvens, Universidade Regional do Cariri, Santana do Cariri, Ceará, Brazil

* These authors contributed equally to this work.

## ABSTRACT

**Background**. A new fossil species of whipscorpion, *Mesoproctus rayoli* **n. sp.**, is described. The specimen originates from the Crato Formation, dating to the Lower Cretaceous (Aptian/Albian) period within the Araripe Sedimentary Basin. This species has been provisionally assigned to *Mesoproctus* Dunlop, 1998, as it represents the sole known Thelyphonida fossil genus discovered in South America and within Araripe Lagerstätte.

**Methods**. The material underwent detailed description and illustration processes. Key diagnostic characters, such as body length, pedipalpal coxae apophysis, the form of the opisthosoma, and the length of leg IV, were meticulously examined. SEM methods were applied in this study.

**Results**. Through the detailed analysis, comparisons and differences to *Mesoproctus rowlandi* Dunlop, 1998 were made possible. Additionally, a well-preserved specimen of the rare camel spider, *Cratosolpuga wunderlichi* Selden, in Selden and Shear, 1996, was identified from the limestones of the Crato Formation. The newly discovered fossil specimen of *Cratosolpuga wunderlichi* suggests two characters not previously described: (i) a segmented tarsomere on leg IV; and (ii) a leg I with one tarsal claw.

Corresponding author
Daniel Lima,
danieljmlima@gmail.com

## INTRODUCTION

Whipscorpions (Thelyphonida/Uropygi) form a distinct group among arachnids, characterized by a large, robust, heavily sclerotized body with a long, annulated, post-abdominal flagellum (Clouse et al., 2017) that presents an overall resemblance to scorpions. Despite their distinctive characteristics, this relatively small arachnid group encompasses 126 extant species (Clouse et al., 2017; World Uropygi Catalog, 2022); however, the fossil record for whipscorpions is exceedingly rare, with only 11 valid species documented in the

**Table 1** Fossil species of Thelyphonida known to date indicating age and place of origin.

| | | |
|---|---|---|
| *Burmathelyphonia prima* Wunderlich, 2015 | Cen | Myanmar |
| *Geralinura britannica* Pocock, 1911 | Duc (=Bash) | England |
| *Geralinura carbonaria* Scudder, 1884 | Mos | USA |
| *Mesoproctus rayoli* **n. sp.** | Apt/Alb | Brazil |
| *Mesoproctus rowlandi* Dunlop, 1998 | Apt/Alb | Brazil |
| *Mesothelyphonus parvus* Cai & Huang, 2017 | Cen | Myanmar |
| *Parageralinura marsiglioi* Selden, Dunlop & Simoneto, 2016 | Kasi–Gzh | Italy |
| *Parageralinura naufraga* (Brauckmann & Koch, 1983) | Penns | Germany |
| *Parageralinura neerlandica* (Laurentiaux-Viera & Laurentiaux, 1961) | Penns | The Netherlands |
| *Parilisthelyphonus bryantae* Knecht et al., 2023 | Mos | USA |
| *Proschizomus petrunkevitchi* Dunlop & Horrocks, 1996 | Duc (=Bash) | England |
| *Prothelyphonus bohemicus* (Kušta, 1884) | Penns | Czech Republic |

**Notes.**
Alb, Albian; Apt, Aptian; Bash, Bashkirian; Cen, Cenomanian; Duc, Duckmantian; Gzh, Gzhelian; Kasi, Kasimovian; Mos, Moscovian; Penns, Pennsylvanian.

**Table 2** Fossil species of Solifugae known to date indicating age and place of origin.

| | | |
|---|---|---|
| *Cratosolpuga wunderlichi* Selden, 1996 | Apt/Alb | Brazil |
| *Cushingia ellenbergeri* Dunlop et al., 2015 | Cen | Myanmar |
| *Eognosippus fahrenheitiana* Dunlop, Erdek & Bartel, 2023 | Lut | Europe |
| *Happlodontus proterus* Poinar & Santiago-Blay, 1989 | Eoc | Dominican Republic |
| *Palaeoblossia groehni* Dunlop, Wunderlich & Poinar, 2004 | Eoc | Europe |
| *Protosolpuga carbonaria* Petrunkevitch, 1913 | Penns | USA |

**Notes.**
Alb, Albian; Apt, Aptian; Cen, Cenomanian; Eoc, Eocene; Lut, Lutetian; Penns, Pennsylvanian.

literature (*Tetlie & Dunlop, 2008*; *Selden, Dunlop & Simonetto, 2016*; *Cai & Huang, 2017*; *Dunlop, Penney & Jekel, 2023*; *Knecht et al., 2023*) (Table 1).

The camel spiders, comprising approximately 1,209 extant species, exhibit unmistakable features such as massive chelicerae and leg-like pedipalps that are held above the ground while the animal is in motion (*Dunlop et al., 2015*; *Beron, 2018*; *Harms & Duperre, 2018*; *World Solifugae Catalog, 2022*). Despite their greater diversity nowadays, the fossil record of camel spiders remains sparse, with only six documented fossil species (Table 2). Remarkably, the fossil history of these small arachnid groups is ancient, with specimens dating back from the Carboniferous period, and species from the Cretaceous, Eocene and Miocene.

The Lower Cretaceous Crato Formation, is a Fossil Konservat Lagerstätte (*Martill & Bechly, 2007*), with remarkable fossil preservation of several groups and delicate structures, such as feathers (*Sayão, Saraiva & Uejima, 2011*), soft tissue of some vertebrate groups (*Field & Martill, 2017*), shrimps (*Alencar et al., 2023*), myriapods (*Wilson, 2003*), hundreds of insect species (*Moura-Júnior, Scheffler & Fernandes, 2018*), and some arachnids (*Santana & Santos Filho, 2021*). One species of whipscorpion and one camel spider are known from the Crato Formation, *Mesoproctus rowlandi* Dunlop, 1998, and *Cratosolpuga*

*wunderlichi* Selden, in *Selden & Shear, 1996*, respectively. *Dunlop & Martill (2002)* reported an additional whipscorpion as *Mesoproctus* sp., a very large specimen not formally described by the authors due to the lack of several characters and poor conditions of the fossil material (*Dunlop, Menon & Selden, 2007*).

Here, in light of the finding of two new very large whipscorpion specimens, we describe a new species, *Mesoproctus rayoli* **n. sp.**, from the Crato Formation, and compare it with *Mesoproctus rowlandi* Dunlop, 1998 and *Mesoproctus* sp. The opportunity is taken to briefly discuss the diagnostic characters of the genus.

Given the discovery of a well-preserved specimen of the camel spider *Cratosolpuga wunderlichi* Selden, in *Selden & Shear, 1996* from the Crato Formation, we present illustrations and describe previously unidentified characters of this species. This also affords us the opportunity to briefly discuss the diagnostic characteristics of the genus.

## Geological setting

The Araripe Sedimentary Basin is located in the interior part of the northeast region of Brazil and includes areas of three different states: Ceará, Pernambuco, and Piauí. The Araripe Basin is about 12,000 km$^2$ and is the largest intratectonic basin in Brazil forming a sedimentary unit including nine geological formations with Devonian, Jurassic, and Cretaceous deposits (*Saraiva et al., 2007*; *Fambrini et al., 2020*). Under the Chapada do Araripe, we found a stratigraphic sequence of ~1,000 m with pre-, syn-, and post-rift phases (*Saraiva et al., 2007*).

The Santana Group (syn-rift phase) is a Cretaceous sedimentary unit subdivided into four formations: Barbalha, Crato, Ipubi, and Romualdo (*Assine, 1994*). Barbalha, Crato, and Ipubi formations are commonly associated with freshwater environments, whereas the Romualdo Formation is associated with lacustrine and transitional marine sediments (*Valença, Neumann & Mabesoone, 2003*; *Fara et al., 2005*).

The Crato Formation is mainly composed of laminated micritic limestones and clay-carbonate rhythmites, with the color of these rocks being bluish in their unweathered state. However, upon weathering, the colors vary from beige to brown and from light gray to bluish gray, which alternate between shales and fine sandstones (*Viana & Neumann, 2002*; *Saraiva et al., 2021*). In the laminated limestone, there are pseudomorphs of salt (halite) and various types of well-preserved fossils (*Viana & Neumann, 2002*; *Martill, 2007*). Outcrops of the Crato Formation are more commonly found in the towns of Nova Olinda and Santana do Cariri, as well as in the quarries of Rio Batateiras, Santa Rita, and Caldas (*Viana & Neumann, 2002*).

The holotype of *Mesoproctus rayoli* **n. sp.** studied here was recovered along with several other fossil specimens in the "Operation Santana Raptor" of the Brazilian Federal Police. The materials recovered by this operation are largely from the Crato Formation, as the famous "Ubirajara jubatus". As this material is from a federal police operation that retrieved smuggled fossils, the precise locality is unknown; however, through the characteristic of the matrix, it is possible to assume its Crato Formation origin. The paratype of *Mesoproctus rayoli* **n. sp.** was donated to the Museu de Paleontologia Plácido Cidade Nuvens (MPPCN) and its origin or any other information about sampling site was not given, although, as is

the case of the holotype, we can assume its Crato Formation origin based on the matrix where the fossil is found.

The Solifugae *Cratosolpuga wunderlichi* Selden, in *Selden & Shear, 1996*, was donated through the "Projeto Força Tarefa" from the Inhumas neighborhood in Santana do Cariri, Ceará, Brazil. This initiative encourages local children to collect and contribute fossils to the MPPCN. Additionally, this project aligns with the museum's campaign, "#lugardefossilenomuseu" which aims to solicit fossil donations from the regional community.

## MATERIALS & METHODS

Descriptions, drawings, and photographs were made using a stereomicroscope Nikon SMZ 745T equipped with camera lucida and a Leica EZ4 W, both with a digital camera attached. Photographs of the whole specimens were made using a Canon G10 camera. The holotype and paratype of *Mesoproctus rayoli* **n. sp.** and *Cratosolpuga wunderlichi* were not coated, and ultraviolet lights were used to identify some of the material's structures. The description is based on both the holotype and paratype of *Mesoproctus rayoli* **n. sp**, and the terminology used in the descriptions mostly followed *Dunlop & Martill (2002)* and *Barrales-Alcalá, Francke & Prendini (2018)*. For the study of *Cratosolpuga wunderlichi*, we mostly used the terminology of *Muma (1951)*, *Shultz (1989)* and *Shultz (1990)*. SEM micrographs were obtained in a SU3500 scanning electron microscope (Hitachi, Tokyo, Japan). The regions of interest were imaged using a SE detector, with accelerating voltages of 20 kV. The fossil material was inserted into the microscope chamber without sample preparation, and the analyses were performed in high vacuum. The specimens of *Mesoproctus rayoli* **n. sp.** could not be analyzed using SEM because their size exceeded the capacity of the microscope chamber. The recent species *Mastigoproctus brasilianus* (*Koch, 1843*) was used in the comparisons. The material is deposited in the MPPCN. Other institutions mentioned include the following: Museum für Naturkunde, Berlin (MB.A); Staatliches Museum für Naturkunde Karlsruhe (SMNK); and Staatliches Museum für Naturkunde Stuttgart (SMNS). Measurements are given in millimeters (mm), and additional abbreviations are L, legs; and L/W, length/width ratio.

The electronic version of this article in portable document format (PDF) will constitute a recognized publication under the International Commission on Zoological Nomenclature (ICZN). Consequently, the new names introduced in the electronic version are deemed officially published under the ICZN solely from the electronic edition. This published work, along with the nomenclatural acts it encompasses, has been duly registered in ZooBank, the web-based registration system for the ICZN. You can access and view the associated information by resolving the ZooBank LSIDs (Life Science Identifiers) through any standard web browser by adding the LSID to the prefix http://zoobank.org/. The LSID for this publication is: urn:lsid:zoobank.org:pub:B4157BE1-FC9E-49F0-A14F-68E838FC241D. The online version of this work is archived and available from the following digital repositories: PeerJ, PubMed Central SCIE and CLOCKSS.

# RESULTS

## Systematic paleontology

Order Thelyphonida *Latreille, 1804*
Thelyphonidae *Lucas, 1835*
Mastigoproctinae *Speijer, 1933*
Genus *Mesoproctus* Dunlop, 1998
*Mesoproctus rayoli* **n. sp.** urn:lsid:zoobank.org:act:1C7BEEFE-795D-4560-9372-B8F17CE7801B
Figs. 1, 2A–2E, 3A–3B, 4, 9

*Mesoproctus* sp.–*Dunlop & Martill, 2002*: 332, figs. 3b, 4b.

**Type material.**—Holotype MPSC A4295, body length 65.9 mm; paratype MPSC A4205, carapace length 28.2 mm.

**Additional specimen.**—Partially complete prosoma MB.A. 1041, carapace length 32.5 mm.

**Type locality.**—Unknown (see Geological setting).

**Stratigraphic unit.**—Crato Formation, Santana Group, Araripe Sedimentary Basin, Brazil.

**Type age.**—Lower Cretaceous (Aptian—Albian).

**Diagnosis.**—Body length including pygidium around 66 mm; pedipalpal coxae apophysis with one terminal tooth, one accessory tooth on the inner margin, outer margin of the apophysis serrated proximally; opisthosoma distinctly oval shaped; leg IV much longer than opisthosoma.

**Description.**—Holotype (Figs. 1, 2A–2E, 3) almost complete, in ventral view; body length (including pygidium) 65.9 mm; carapace length 25.7 mm, width 14.3 mm (L/W 1.8 mm). Y-shaped anterior sternum between second pair of leg coxae (Fig. 2B). Posterior sternum not well preserved.

Pedipalps massive, folded, with coxae, femur, patella and tibia partially preserved (Figs. 2A, 3A); trochanter not well preserved. Pedipalpal coxae apophysis short, rounded, with one terminal tooth, and 1 accessory tooth on the inner margin, outer margin serrated proximally (Figs. 2A, 2C, 3B). Trochanter poorly preserved, with 1 spine visible on the right side (Fig. 2C) and 3 spines on the left side (Fig. 3A). Femur twice as long than wide; patella 1.4 longer than wide; patellar apophysis well developed. Tibia 1.6 longer than wide (excluding fixed finger and tibia apophysis) (Fig. 2D); fixed finger and basitarsus well developed.

Legs mostly preserved flattened (L4 > L2 = L3), prolateral face uppermost, partially complete, distal end not well preserved, telotarsus apparently not divided. Antenniform leg I not preserved. Legs II and III about same size, robust femora, patellae and tarsi ventral corner with stiff setae (clearly preserved on left side of the specimen), right leg II with pair of small claws. Leg IV longer than II and III, well exceeding pygidium.

Opisthosoma elongate; longitudinally oval, length 37.2 mm, width 16.5 mm (L/W ratio 2.2) (Figs. 1, 2E); first and last three segments (pygidium) poorly preserved; sternites

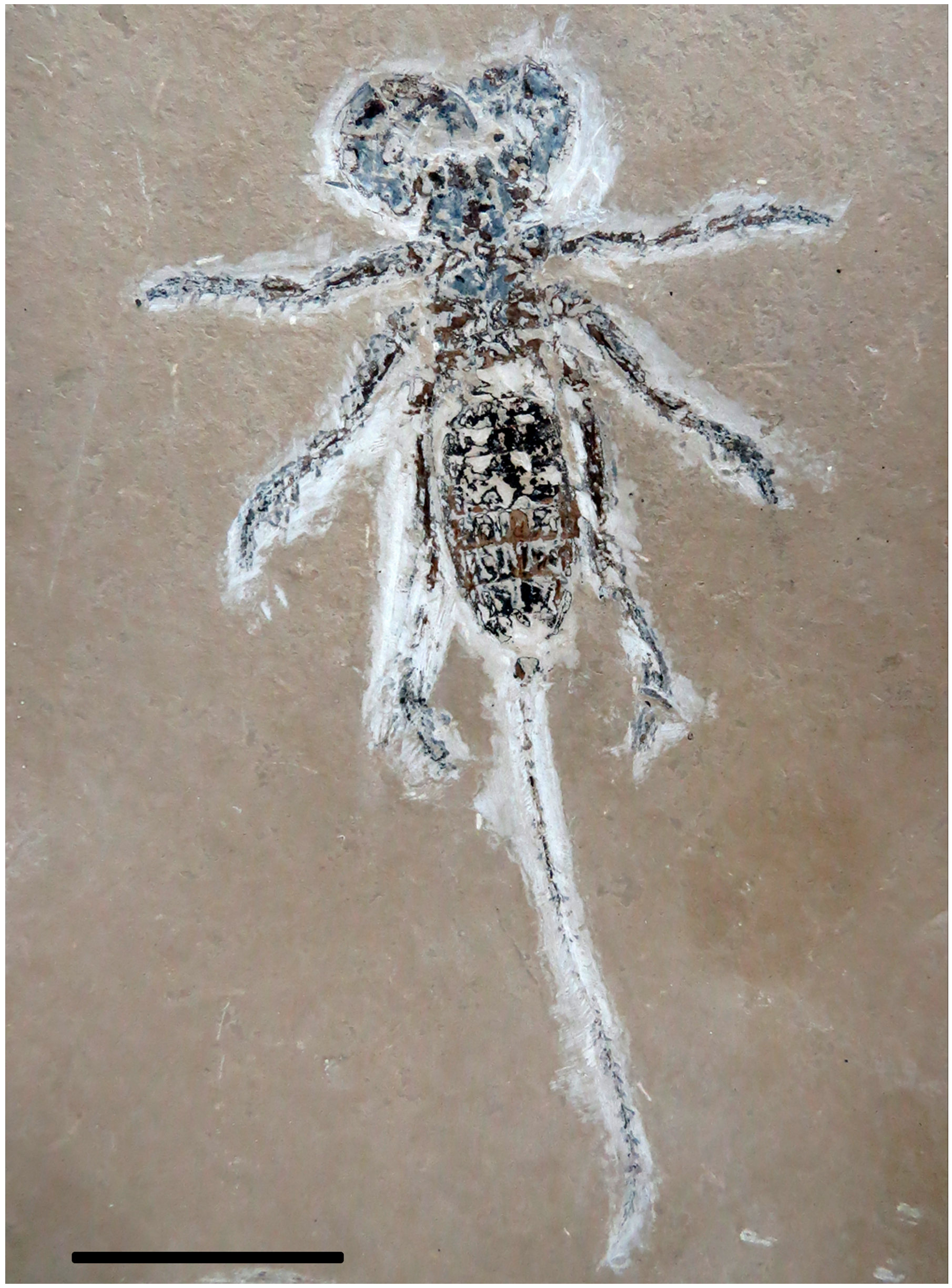

**Figure 1** ***Mesoproctus rayoli*** **n. sp. Holotype MPSC A4295.** Ventral view. Scale: 30 mm.

II–IX partially preserved; telson flagelliform, length 72.5 mm, width 1.1 mm; articles approximately twice as long as wide.

Paratype (Fig. 4) partially preserved in dorsal view with only first three segments of opisthosoma preserved; carapace elongate, densely granulated; granules uniformly distributed; length 28.2 mm, width 16.6 mm (L/W ratio 1.7 mm).

**Etymology.**—The specific epithet is in honor of Rafael Ribeiro Rayol, Brazilian federal attorney, who helped, along with the Brazilian Federal Police, in the retrieve of the material from the "Operation Santana Raptor".

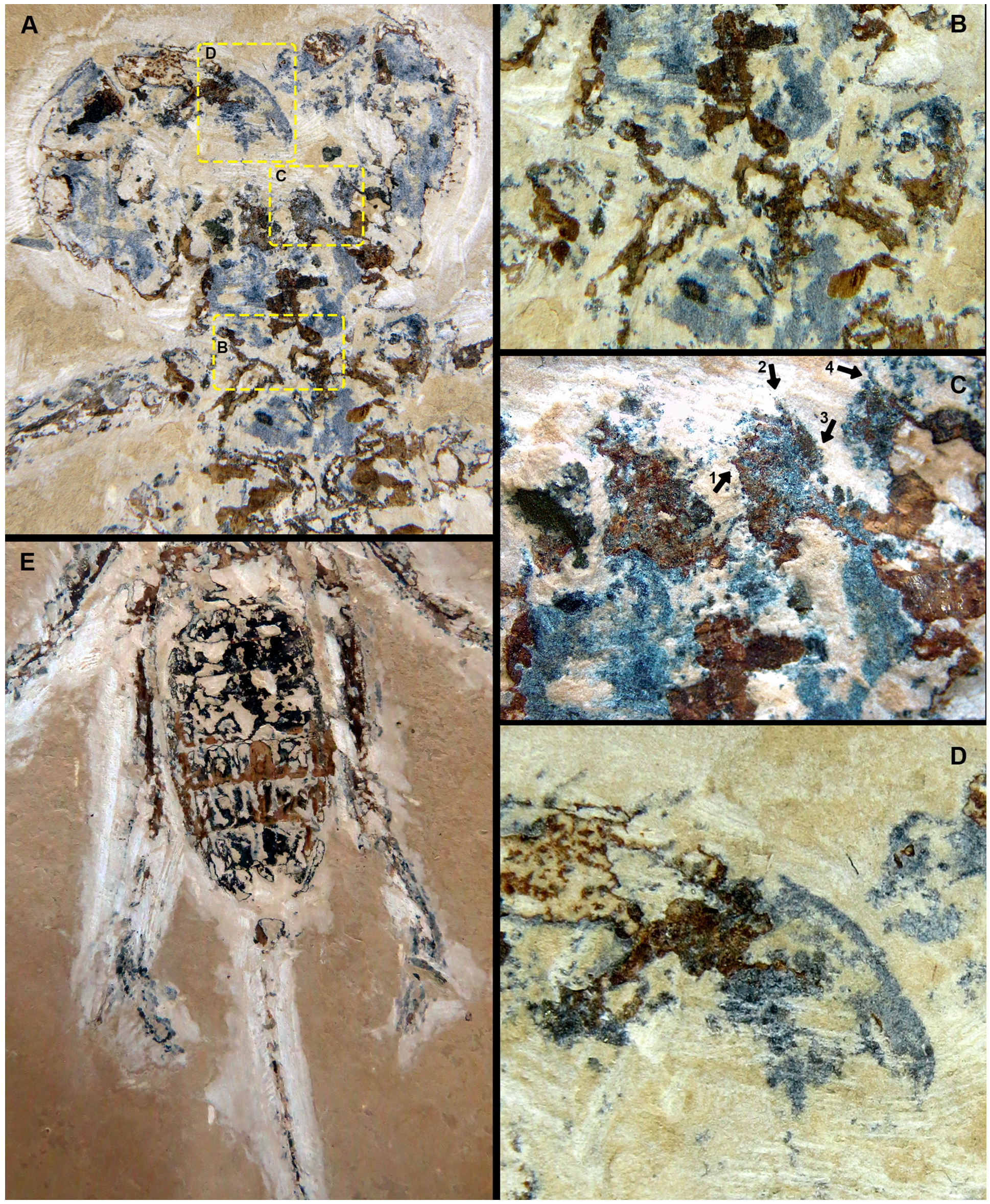

**Figure 2** ***Mesoproctus rayoli* n. sp. Holotype MPSC A4295, ventral view.** (A) Pedipalp and anterior sternum. Yellow rectangles: detail of (B), (C), and (D). (B) Y-shaped anterior sternum between second pair of leg coxae. (C) Left pedipalpal coxae apophysis and trochanter. Arrow 1: inner margin accessory tooth; arrow 2: terminal tooth; arrow 3: outer margin serrated proximally; arrow 4: trochanter spine. (D) Fixed finger and basitarsus. (E) Opisthosoma and legs IV.

**Remarks.**—When adding new characters to the description of *Mesoproctus rowlandi* Dunlop, 1998, *Dunlop & Martill (2002)* proposed that a larger specimen (MB.A. 1041) examined might represent a distinct species. This suggestion primarily stemmed from the significant size difference between *Mesoproctus* sp. and *M. rowlandi*. Alternatively, they also suggested that this size discrepancy might be attributed to the differing ontogenetic stages of the specimens, with the smaller one potentially being a juvenile. In the light of the

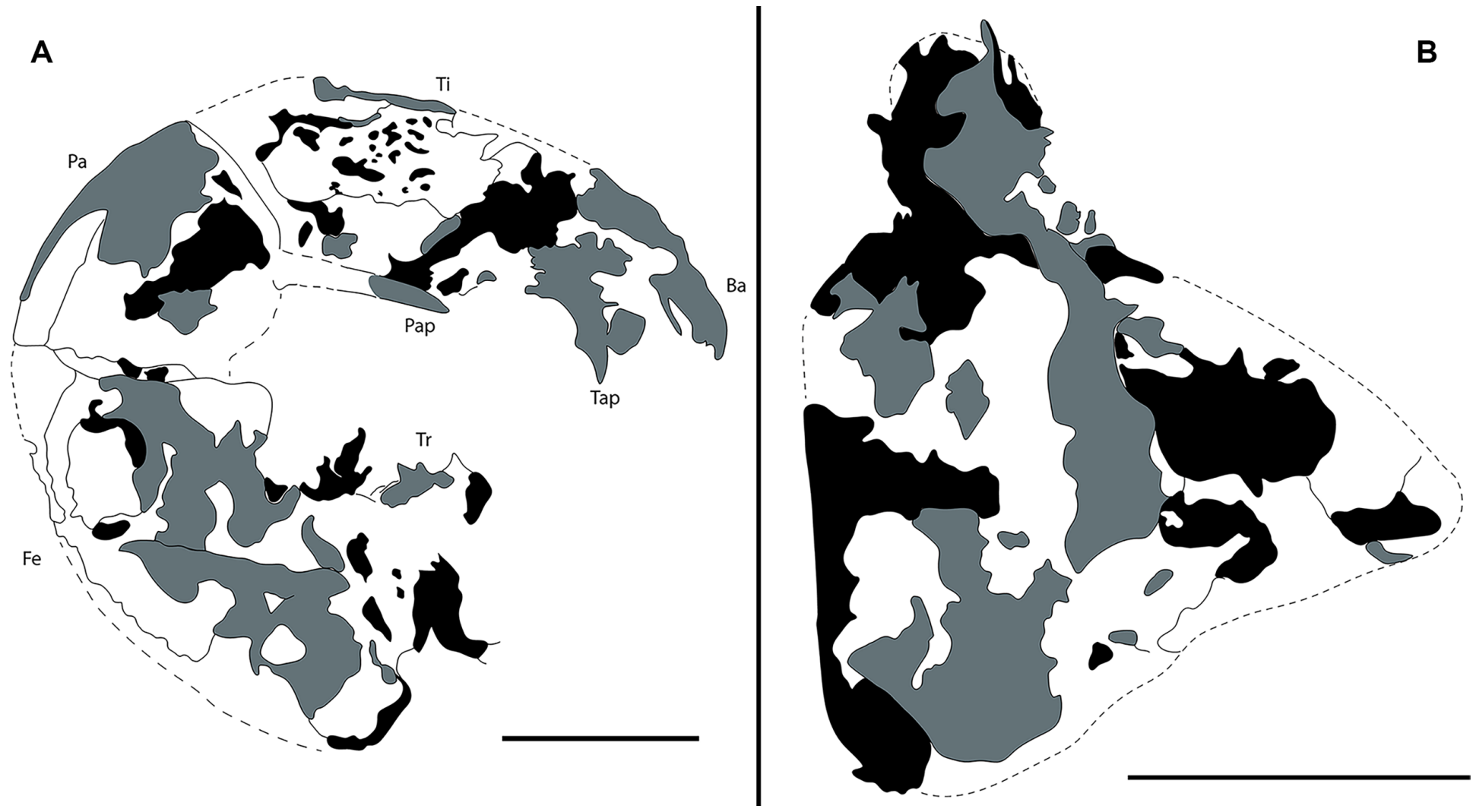

**Figure 3** **Camera lucida drawings of Mesoproctus rayoli n. sp. holotype MPSC A4295.** (A) Right pedipalp, ventral view. (B) Left pedipalp coxa, ventral view. Ba, basitarsus; Fe, femur; Pap, patella apophysis; Tap, tibia apophysis; Ti, tibia; Tr, trochanter. Black, areas of dark mineralization; Gray, areas of lighter mineralization; White, areas of background matrix; dotted lines, represent not preserved margins. Line drawing by *D. Lima*. Scale: 5 mm.

new material studied, here we found characters to differentiate *Mesoproctus rayoli* **n. sp.** from *Mesoproctus rowlandi* Dunlop, 1998 and include *Mesoproctus* sp. of *Dunlop & Martill (2002)* in the synonym of the new species.

The size of *Mesoproctus rayoli* **n. sp.** (holotype 65.9 mm length) is impressive, without comparison in the fossil record. Among the whipscorpions, the only group with a similar size is the extant *Mastigoproctus* Pocock, 1894, with species that can reach as far as 73 mm in length including the pygidium (*Barrales-Alcalá, Francke & Prendini, 2018*). Within the known specimens of *Mesoproctus rowlandi* Dunlop, 1998, the type specimen has 25 mm (*Dunlop, 1998*), and additional specimens with 16.8 and 17.1 mm in length (without pygidium) were described by *Dunlop & Martill (2002)*. *Mesoproctus* sp. from *Dunlop & Martill (2002)*, which we consider here as a junior synonym of *Mesoproctus rayoli* **n. sp.,** had a carapace length of 32.5 mm, being even larger than the holotype. It is true that in many species, mature specimens can be distinguished from their immature counterparts based on size; however, as observed in the recent species *Mastigoproctus giganteus*, ornamentation of the pedipalp remains somewhat constant during ontogeny (*Weygoldt, 1971*). Thus, *Mesoproctus rayoli* **n. sp.** differs from *Mesoproctus rowlandi* (Dunlop, 1998) by the pedipalpal coxae apophysis having one terminal tooth, one accessory tooth on the inner margin, and the outer margin of the apophysis serrated proximally (Figs. 2A, 2C, 3B) (*vs.* endites of coxa of pedipalps without ornamentation in *M. rowlandi*). *Mesoproctus rayoli* **n. sp.** also differs from *M. rowlandi* in the proportions of the opisthosoma, distinctly oval shaped in *M. rayoli* **n. sp.** (Figs. 1, 2E) and considerably cylindrical in *M. rowlandi* (*Dunlop, 1998*: 292, fig. 1; *Dunlop & Martill, 2002*: 327, fig. 2B; 332, fig. 4A). For instance,

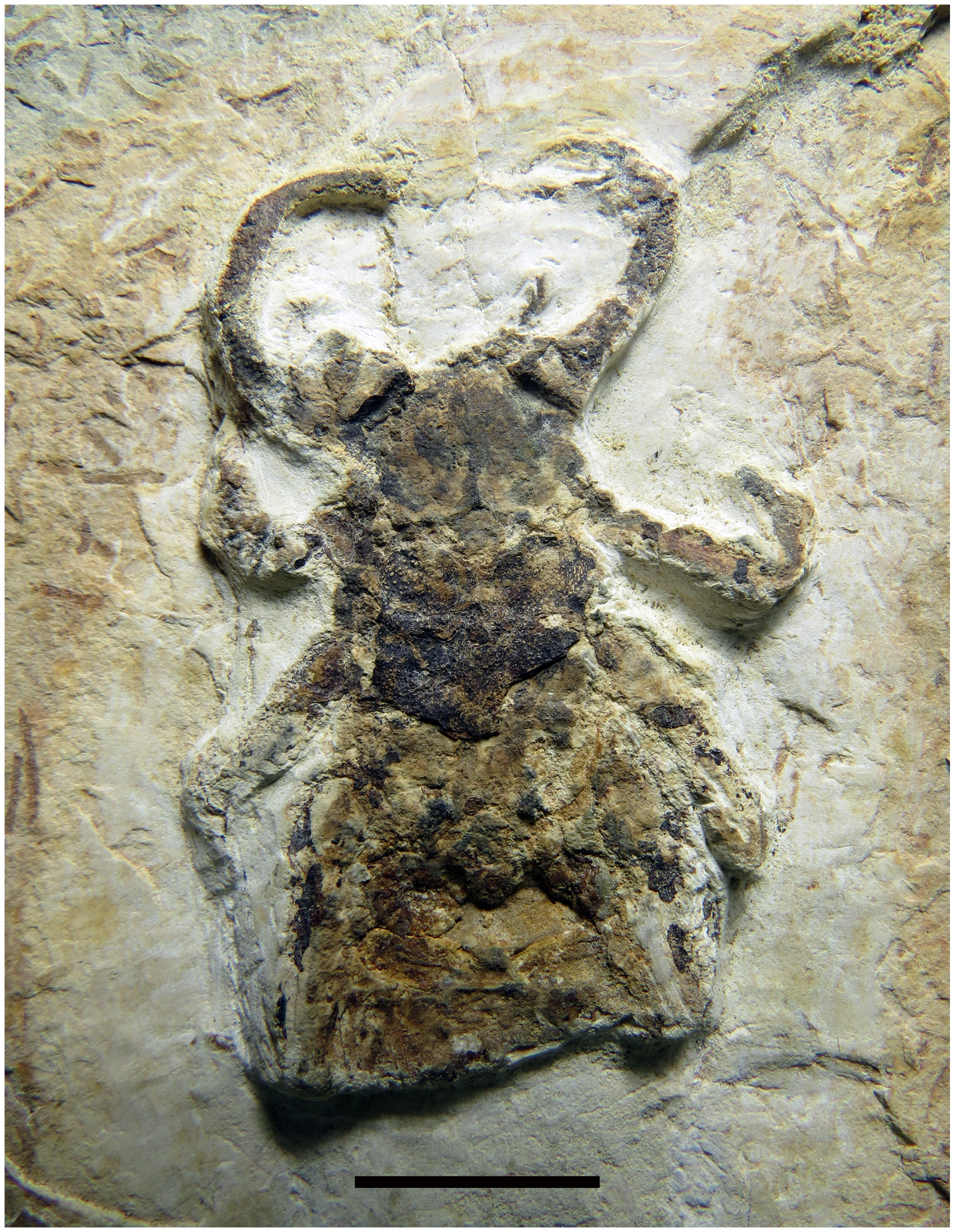

**Figure 4** ***Mesoproctus rayoli* n. sp. Paratype MPSC A4205.** Dorsal view. Scale: 10 mm.

the L/W ratio of the opisthosoma sixth segment is 4.0 in *Mesoproctus rayoli* **n. sp.** (Fig. 2E), and the L/W ratio of the opisthosoma sixth segment in *M. rowlandi* is 2.1. Also, leg IV is much longer than the opisthosoma in *M. rayoli* **n. sp.** (Figs. 1, 2E), whereas the fourth pair of legs were just reaching (MB.A 975) or slightly overreaching the pygidium in *M. rowlandi* (holotype) (*Dunlop, 1998*: 292, fig. 1; *Dunlop & Martill, 2002*: 327, fig. 2B; 332, fig. 4A).

Order Solifugae *Sundevall, 1883*
Family Ceromidae? *Roewer, 1934*

**Remarks.**–When describing *Cratosolpuga wunderlichi*, Selden, in *Selden & Shear, 1996*, attributed the new taxon to Ceromidae based on the morphology of the flagellum, the

propeltidium shape with distinct lateral lobes, the eye tubercle with anteriorly directed spines, the spine pattern of legs, and the presence of long tarsal claws on legs III and IV (see also *Punzo, 1998*). The specimen we report here aligns with all of the characteristics described by Selden (*Selden & Shear, 1996*). However, the new material suggests the possibility of a segmented tarsomere on leg IV and leg I with one tarsal claw. Such features were neither observed in the holotype by *Selden & Shear (1996)* nor confirmed by *Dunlop & Martill (2004)*. In our analysis, we identified what appears to be segmentation on the tarsomere of leg IV and at least one leg I tarsal claw. Although these characteristics aid in family definition (*Muma, 1976*), the potential number of tarsomeres observed aligns with the criteria set for Ceromidae as per *Selden & Shear (1996)*.

Genus *Cratosolpuga* Selden, in *Selden & Shear, 1996*
**Diagnosis** (after Selden, in *Selden & Shear, 1996*).–Male cheliceral flagellum attached to dorsomedial side of chelicera near base of fixed finger, consisting of globose base and styliform whip extending directly backwards to base of chelicera, stiff membrane partly enclosing flagellum base and running length of flagellum forming a narrow gutter.

*Cratosolpuga wunderlichi* Selden, in *Selden & Shear, 1996*
Figs. 5, 6A–6F, 7A–7F, 8A–8D, 9
*Cratosolpuga wunderlichi* Selden, in *Selden & Shear, 1996*: 584, 588–595, 601–603, pl. 1, figs. 2–4, pl. 2, text-figs. 1, 3–6 a–g.
*Cratosolpuga wunderlichi*— *Punzo, 1998*: 213–214, figs. 7.7–7.10; *Dunlop & Martill, 2002*: 325; *Harvey, 2002*: 366; *Harvey, 2003*: 212; *Dunlop & Martill, 2004*: 145, figs. 1–8; *Dunlop, Menon & Selden, 2007*: 120, figs. 9.7, 9.8; *Dunlop & Penney, 2012*: 72, fig. 46.
*Cratosolpuga* [*wunderlichi*]—(*Dunlop, 1996*): 85, fig. 5.
**Type material**.–Holotype (nº Sol 1), part only, deposited in the private collection of J. Wunderlich, Germany.
**Additional specimens**.–SMNK 1268 PAL 1, presumed juvenile, total length 5.8 mm; MB.A. 1087, total length 20.3 mm; MB.A.1088, total length 14.4 mm; SMNS 65417, total length 14.7 mm; SMNS 54418, total length 23.1 mm.
**Stratigraphic unit**.–Crato Formation, Santana Group, Araripe Sedimentary Basin, Brazil.
**Type age**.–Lower Cretaceous (Aptian–Albian).
**Material examined.**–MPSC A6696, total length 16.5 mm, probably an adult male; collected from Santana do Cariri, Ceará, Brazil.
**Remarks**.–Measurements of the studied specimen: Total body length (including chelicerae): 16.48 mm (Fig. 5); prosoma (with chelicerae): 4.97 mm; prosoma (without chelicerae): 4.06 mm; opisthosoma: 9.27 mm; chelicera: 0.91 mm. Total length of leg 1: 8.55 mm; femur: 1.03 mm, patella: 2.21 mm, tibia: 2.21 mm, basitarsus: 1.5 mm, telotarsus: 1.6 mm. Total length of leg 2: 5.18 mm; femur: 0.95 mm, patella: 1.22 mm, tibia: 1.46 mm, basitarsus: 0.97 mm, telotarsus: 0.58 mm. Total length of leg 3: 5.97 mm; femur: 1.01 mm, patella: 1.43 mm, tibia: 1.58 mm, basitarsus: 1.23 mm, telotarsus: 0.72 mm. Total length of leg 4 (exposed part): 5.38 mm. Protopetidium: length = 3.26 mm, width = 2.07 mm.

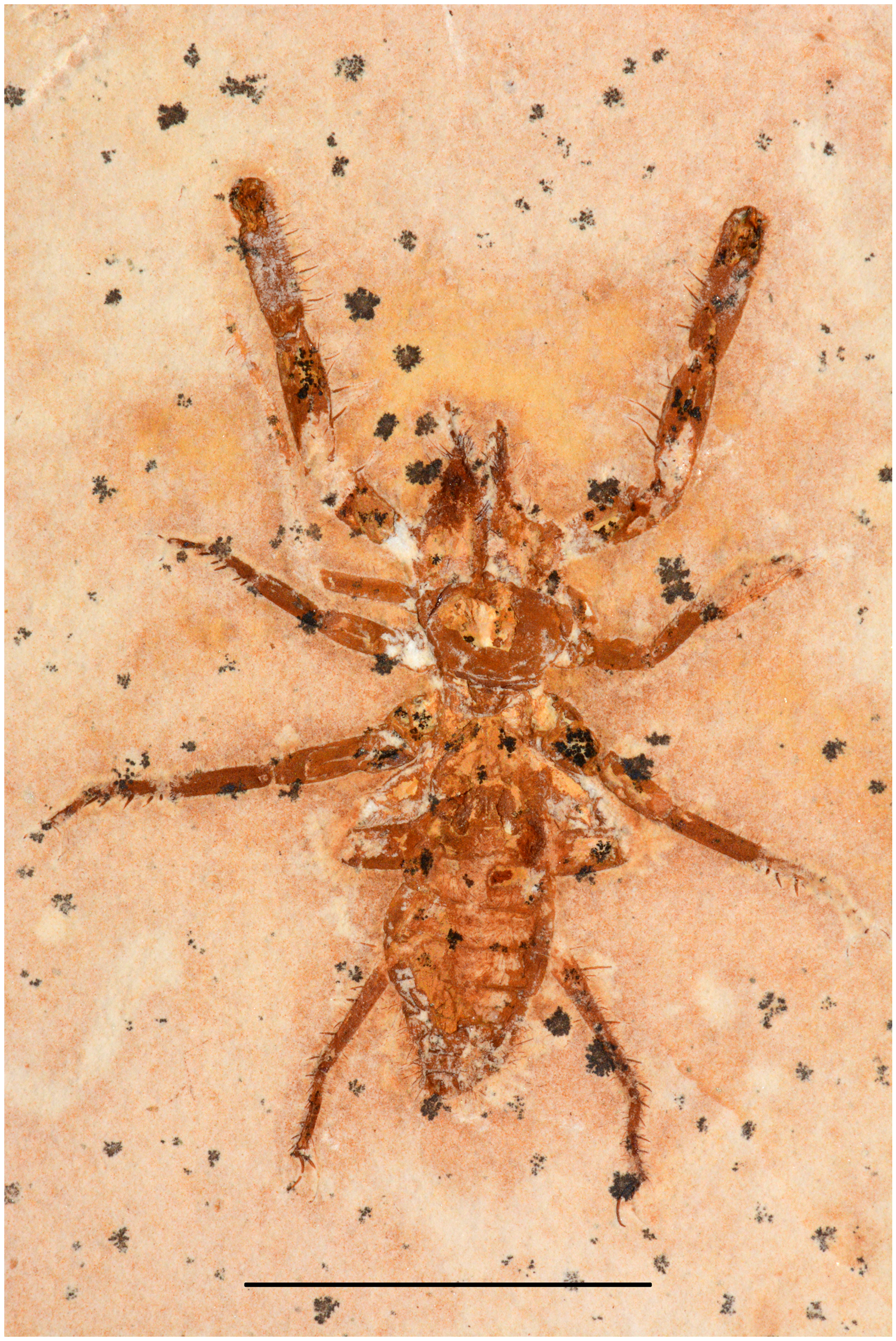

**Figure 5** ***Cratosolpuga wunderlichi*** **Selden, in *Selden & Shear, 1996*. MPSC A6696.** Dorsal view. Scale: 10 mm.

The specimen studied here presents what is probably the flagellum of the chelicera, a structure present only in adult males (Figs. 6A, 6B). Although this structure is not exceptionally preserved, it seems to be a styliform flagellum with a bulbous base. The prosoma has an exterior lobe clearly separated from the main part of the propeltidium (Fig. 7A), and both features closely mirror the observations made by *Selden & Shear (1996)* for ceromids.

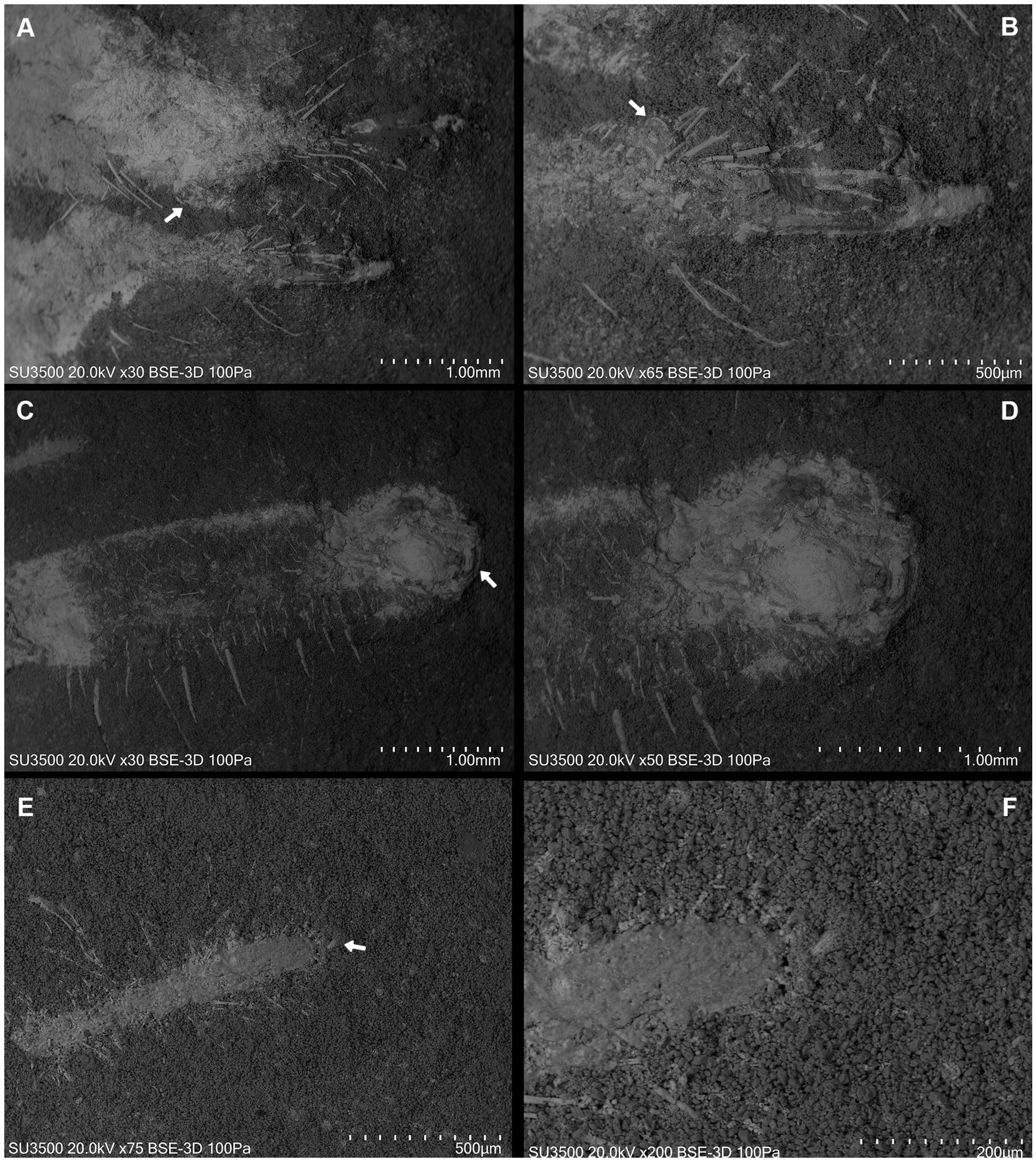

**Figure 6** ***Cratosolpuga wunderlichi*** **Selden, in (*Selden & Shear, 1996*). MPSC A6696.** Dorsal view SEM images. (A) Right and left chelicerae. White arrow: flagellum. (B) Right chelicerae. White arrow: flagellum. (C) Left pedipalp tarsus and tibia. White arrow: putative sucker organ. (D) Inverted "C" sucker organ detail. (E) Left leg I anterior part. White arrow: tarsal claw. (F) Detail of (E).

The morphology of the pedipalp, including its shape and spinulation, aligns well with the descriptions given by *Selden & Shear (1996)*. In particular, the tarsus is short and bulbous, with the tibia-tarsus joint being distinctive (Figs. 6C, 6D, 7B). Moreover, an inverted "C" structure located toward the anterior portion of the tarsus could represent the sucker organ (Figs. 6C, 6D).

Leg I exhibits a distinct tarsal claw (Figs. 6E, 6F, 7C). The tarsal segment formula (leg II, III, and IV) identified for *Cratosolpuga* in this study is 1, 1, 2(3), presenting a slight difference from the 1, 1, 1 formula detailed by *Selden & Shear (1996)* (Figs. 7F, 8A, 8B). It is noteworthy that the added tarsal segmentation observed in leg IV is consistent with

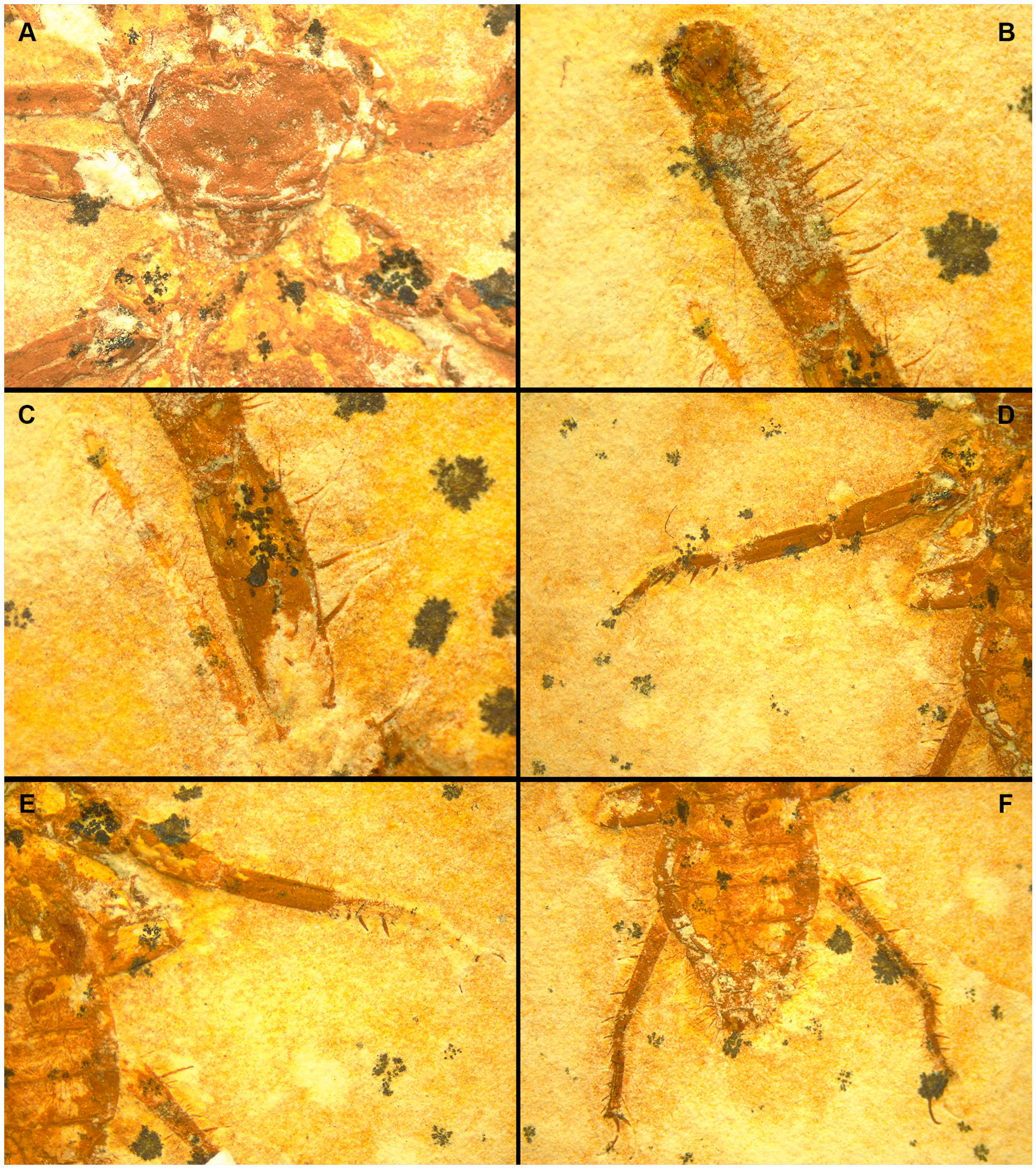

**Figure 7** ***Cratosolpuga wunderlichi*** **Selden, in (*Selden & Shear, 1996*). MPSC A6696.** Dorsal view. (A) Prosoma without chelicerae. (B) Distal part of left pedipalp and leg I. (C) Anterior leg I detail. (D) Left and (E) right leg III. (F) Opisthosoma and leg IV.

patterns in Ceromidae (1, 2, 2), though it differs from the configuration seen in leg III (Figs. 7B, 7E).

Ceromids are typified by a unique pulvillus located on a small apotele of legs II–IV. However, this characteristic was not highlighted by *Selden & Shear (1996)*. In the specimen we studied, a clear pulvillus is visible on the left leg IV (Fig. 8C). It is essential to emphasize that the presence of a pulvillus is not exclusively diagnostic for Ceromidae, as mentioned by *Selden & Shear (1996)*. Additionally, the dorsal positioning of the anus on the anal segment is likely due to taphonomic processes (Fig. 8D), considering that it is ventrally located in Rhagodidae, and is terminally placed in other families (*Harvey, 2003*).

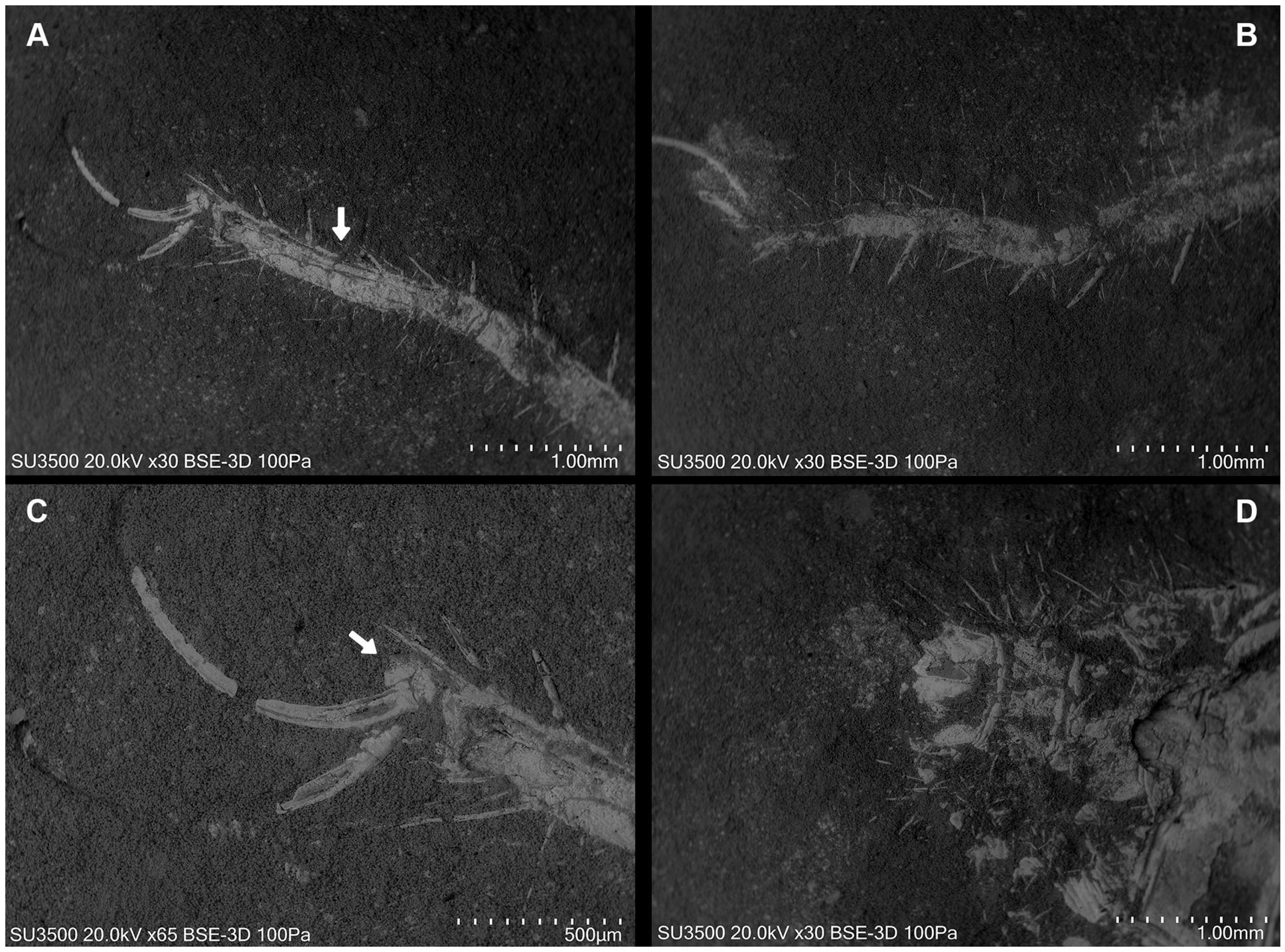

**Figure 8 *Cratosolpuga wunderlichi* Selden, in *Selden & Shear, 1996*. MPSC A6696.** Dorsal view. (A, B) Left and right leg IV tarsomere and tarsal claws. White arrow: additional tarsal joint. (C) Right leg IV tarsal claws. White arrow: pulvillus. (D) Anal segment with terminal anus dorsally displaced.

## DISCUSSION

The discovery of the new fossil whipscorpion, *Mesoproctus rayoli* **n. sp.**, and the rare specimen of *Cratosolpuga wunderlichi* provides valuable insights into the ancient arachnid fauna of the Lower Cretaceous. The description of a second *Mesoproctus* species from the Crato fossil beds, along with the presence of xerophilic arthropods such as camel spiders, scorpions, and insects, lends support to the hypothesis that the Crato Formation was deposited in an arid paleoenvironment (*Grimaldi & Maisey, 1990*; *Martill, 1993*) (Fig. 9).

The occurrence of several gnetophytes, including *Cratonia cotyledon Rydin, Mohr & Friis, 2003*, *Priscowelwitschia austroamericana Dilcher et al., 2005*, *Welwitschiophyllum brasiliense Dilcher et al., 2005*, and *Arlenea delicata Ribeiro et al., in press*, suggests arid environmental conditions. Additionally, salt pseudomorphs, prevalent gypsum beds in adjacent sediments, and the taphonomy of spiders all point toward hypersaline water conditions and a surrounding arid landscape (*Downen, Selden & Hasiotis, 2016*).

However, beyond their scientific contributions, these discoveries underscore the significance of community and organizational involvement in paleontological advancements. The participation of initiatives like the "Operation Santana Raptor" by the Brazilian Federal Police highlights the importance of interdisciplinary collaborations in reclaiming valuable fossil specimens that may have been transported from Brazil. Meanwhile, the donation of *Cratosolpuga wunderlichi* through the "Projeto Força Tarefa"

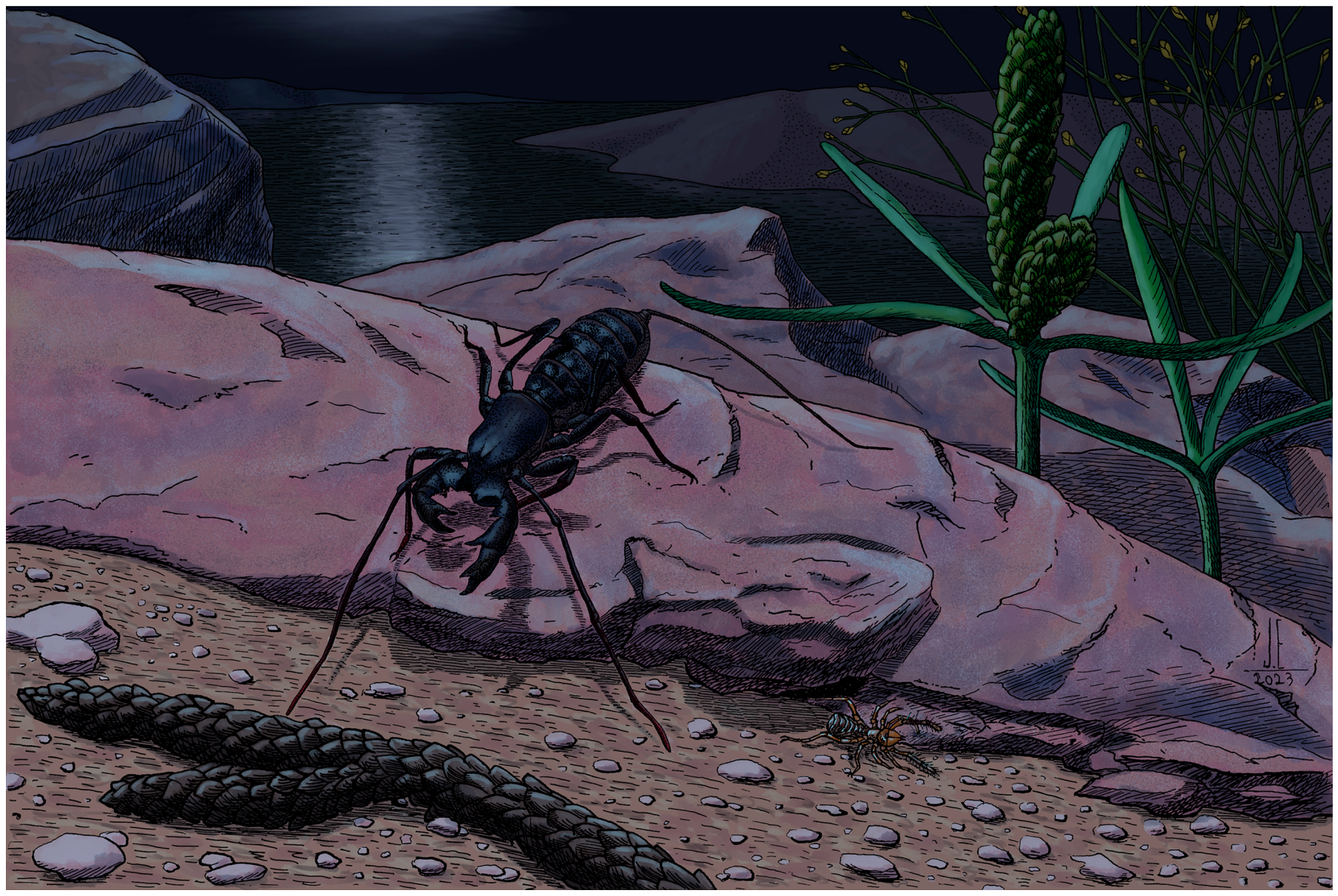

**Figure 9 Reconstruction of *Mesoproctus rayoli* n. sp. and *Cratosolpuga wunderlichi* Selden, in *Selden & Shear, 1996* in their paleoenvironment.** Illustration by the paleoartist João Eudes.

serves as a testament to community-based contributions. Such endeavors, particularly those engaging younger participants, can be pivotal for future discoveries and fostering a culture of conservation and scientific curiosity.

Notably, the "#lugardefossilenomuseu" campaign by the MPPCN exemplifies the power of community mobilization. Encouraging regional communities to actively participate in fossil discovery and donation can significantly enhance the museum's collection while simultaneously promoting awareness and education about regional paleontological heritage.

## ACKNOWLEDGEMENTS

We are grateful to Carlos Eduardo de Souza of the Memorial Beata Benigna for leading the "Projeto Força Tarefa". Their generous donation of the *Cratosolpuga wunderlichi* specimen made this study possible. We also thank Naiara Cipriano Oliveira (URCA) for her assistance with the SEM images, and to José Lúcio e Silva for his contributions to the paleontological collection from the MPPCN. We are deeply appreciative of the comprehensive reviews and highly constructive comments provided by the reviewers, Jason Dunlop (Museum für Naturkunde) and Paul Selden (University of Kansas).

## Funding

This work was supported by the Fundação Cearense de Apoio ao Desenvolvimento Científico e Tecnológico (FUNCAP) (grants #PV1-0187-00019.01.00/21, #BP5-0197-00056.01.00/22 to Daniel Lima; #BP3-0139-00166.01.00/18, #BP4-00172-00173.01.00/20 to Allysson P Pinheiro; and #PV1-0187-00033.01.00/21 and #6647309/2017 to William Santana); the Conselho Nacional de Desenvolvimento Científico e Tecnológico (CNPq) (PQ2 #315185/2020-1 to William Santana); the Coordenação de Aperfeiçoamento de Pessoal de Nível Superior—Brasil (CAPES)—Finance Code 001 (fellowship # 88887.169169/2018-00 to William Santana and grant Proequipamentos #775705/2012 to Allysson P Pinheiro); and the Financiadora de Estudos e Projetos (FINEP) (grant # 1015/13). The funders had no role in study design, data collection and analysis, decision to publish, or preparation of the manuscript.

## Grant Disclosures

The following grant information was disclosed by the authors:
The Fundação Cearense de Apoio ao Desenvolvimento Científico e Tecnológico (FUNCAP): #PV1-0187-00019.01.00/21, #BP3-0139-00166.01.00/18, #BP4-00172-00173.01.00/20, #PV1-0187-00033.01.00/21, #6647309/2017, #BP5-0197-00056.01.00/22.
The Conselho Nacional de Desenvolvimento Científico e Tecnológico (CNPq): PQ2 # 315185/2020-1.
The Coordenação de Aperfeiçoamento de Pessoal de Nível Superior—Brasil (CAPES)—Finance Code 001: #88887.169169/2018-00, #775705/2012.
The Financiadora de Estudos e Projetos (FINEP): #1015/13.

## Competing Interests

The authors declare there are no competing interests.

## Author Contributions

- William Santana conceived and designed the experiments, performed the experiments, analyzed the data, prepared figures and/or tables, authored or reviewed drafts of the article, and approved the final draft.
- Allysson P. Pinheiro performed the experiments, analyzed the data, authored or reviewed drafts of the article, and approved the final draft.
- Thiago Andrade Silva performed the experiments, analyzed the data, prepared figures and/or tables, and approved the final draft.
- Daniel Lima conceived and designed the experiments, performed the experiments, analyzed the data, prepared figures and/or tables, authored or reviewed drafts of the article, and approved the final draft.

## Data Availability

All specimens are deposited at Museu de Paleontologia Plácido Cidade Nuvens: MPSC A4295, MPSC A4205, and MPSC A6696.

## New Species Registration

The following information was supplied regarding the registration of a newly described species:

Publication LSID: urn:lsid:zoobank.org:pub:B4157BE1-FC9E-49F0-A14F-68E838FC241D

Mesoproctus rayoli n. sp. LSID: urn:lsid:zoobank.org:act:1C7BEEFE-795D-4560-9372-B8F17CE7801B

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
