# Peer review of "Description of a new fossil Thelyphonida (Arachnida, Uropygi) and further record of Cratosolpuga wunderlichi Selden, in Selden and Shear, 1996 (Arachnida, Solifugae) from Crato Formation (Aptian/Albian), Araripe Basin, Brazil"

_PeerJ, doi:10.7717/peerj.16670_

## Round 0.1 · original submission · Minor Revisions

Thank you very much for submitting the manuscript. Please take into account all reviewers' comments and suggestions.

**Language Note:** The review process has identified that the English language must be improved. PeerJ can provide language editing services - please contact us at copyediting@peerj.com for pricing (be sure to provide your manuscript number and title). Alternatively, you should make your own arrangements to improve the language quality and provide details in your response letter. – PeerJ Staff

·

Basic reporting

This is a nice contribution describing a new speces of fossil whip scorpion and a new specimen of a fossil camel spider, both from the Cretaceous Crato Formation of Brazil. As the authors correctly note, fossils of both groups are rare and new discoveries merit publication in a journal such as Peer J. The manuscript is generally well written and supported by good photographs. The references are comprehensive and I can only suggest to add a recently published paper on Carboniferous whip scorpions here. The two tables are useful, but they need some revision to standardise the geological ages being used here which are currently inconsistent.

Overall I can recomend acceptance subject to MINOR REVISION and most of my comments are small lingustic corrections.

Experimental design

The manuscript does not include an experiment per se, but the fossils have been described to a high standard and the methods used for photographing the material (including scaning electron microscopy) are appropriate and have yieldied good images. Accompanying intertretative drawings would have been nice, but are not vital. The systematic palaeontology appears correct and the specimens are properly described with a Zoobank registration for the new species. The fossils (including the types of the new species) are now curated in a recognised pubic collection: the Museu de Paleontologia Placido Cidade Nuvens. The history of the specimens being recovered from illegal collecting as well as from outreach activities with the public (school children) is especially interesting.

Validity of the findings

The results appear to be valid and I think a new whip scorpion species is justified, although perhaps a little more discussion to exclude the possibility that there are different ontogenetic stages would be useful. The relevance of the new material is briefly discussed and I think this makes a nice, short contribution to our understanding of the arachnid fossil record.

Additional comments

Suggested minor corrections.


ABSTRACT

line 26: better "...from material collected..."

line 37: better "...differentiation to Mesoproctus..." ?

line 30: maybe add to the abstract the fact that the new camel spider fossil offers two novel characters for the species: i.e. segmented leg IV tarsomere and a leg 1 claw.


INTRODUCTION

line 43: Whipscorpions form a distinct..."

line 45: better "...resemblance to scorpions..."

line 46: why not cite the currrent species count (126) from the World Uropygi Catalog (you do this for the camel spider): https://wac.nmbe.ch/order/uropygi/7

line 55: "...with specimens dating back from..."

line 57: better "The Lower Cretcaeous Crato Formation..."

line 61: Moura-Junior et al. (2018) is not cited in the reference list.

line 62: "One species of whipscorpion and one camel spider are..."

line 64: "An additional whipscorpion was reported..." [past tense]

line 84: "...is a Cretaceous sedimentary subdivided..." ? [a word seems to be missing: system or unit?]

line 91: Newman (with w) or Neumann (with u) as in the references?

line 94: "...are more commonly found in the..."

line 97: better "...other fossil specimens..."

line 98: better "...are largely from the Crato..."

line 100: "...locality is unknown;..."


MATERIALS AND METHODS

line 128: Please add the SMNK to the list of other institutions (see line 250), i.e. the "Staatliches Museum für Naturkunde Karlsruhe".


RESULTS

lines 155/157: I'm not sure if you can use Crato Formation as both the locality and the stratigraphic unit. Please check this as it seemed a bit unusual.

lines 189-215: I think the new species is justified, but as I am sure you relase size is a problematic charater to use and I wonder if the case for this would be strengthened by making more reference to the systematics/ontogeny of living whip scorpions. For example, are the pedipalpal coxal apophyses used in the taxonomy of living species and is there any evidece that this character is stable across both juvenile and adult instars?

I recall Peter Weygoldt wrote about development in whip scorpions. Perhaps its worth checking:

Weygoldt P. 1971. Notes on the life history and reproductive biology of the giant whip scorpion, Mastigoproctus giganteus (Uropygi, Thelyphonidae) from Florida. Journal of Zoology, London 164:137–147.

I would also be a little bit cautious about using opisthosomal proportions in the diagnosis of fossils as these may be influenced by compression or whether the specimen was well-fed or gravid prior to preservation.

line 203: You should probably clarify that the Mesoproctus sp. specimen had a carapace length of 32.5 mm (not its full body length).

line 228: "...these characteristics aid in family..."

line 247: probably better to say "...in the private collection of J. Wunderlich..."

line 254: your definition of the type locality here is different to the one you use on lines 155-157, even though the fossils are from the same Crato Formation. Please be consistent.

line 257 "collected from" better than "sampled at" ?

line 277: better "Leg I exhibiting..." [delete The]


DISCUSSION

lines 293-294: this sentence doesn't seem to make sense. You refer to "a second ceromid whipscorpion", but I presume you mean "a second ceromid camel spider", but then you mention comparative xerophilic arthropods where you list solifuges (an alternative name for camel spiders). Please check this section again.


REFRERENCES

line 368: Fambrini etl. 2020 should come alphabetically before Fara et al. 2005.

line 426: "...phylogeny of Arachnida." [not Araghnida]

line 437: again Newmann or Neumann?


FIGURE CAPTIONS

lines 467-468: I think you can say "leg" not "Leg" here.

line 472: I think it should be "pulvillus" as you only have one and "pulvilli" is plural.


TABLE 1

I think you can add your new species to Table 1 as "Mesoproctus rayoli n. sp.".

You may also want to add data from a two new papers, one of which has some new Carboniferous whip scorpions species, which recently became available online:

Garwood, R. J. & Dunlop, J. A. 2023. X-ray microtomography of the late Carboniferous whip scorpions (Arachnida, Thelyphonida) Geralinura britannica and Proschizomus petrunkevitchi, Journal of Systematic Palaeontology, 21:1, DOI: 10.1080/14772019.2023.2180450

https://www.tandfonline.com/doi/full/10.1080/14772019.2023.2180450

Knecht, R.J., Benner, J. S., Dunlop, J. A. and Renczkowski, M. D. 2023. The largest Palaeozoic whip scorpion and the smallest (Arachnida: Uropygi: Thelyphonida); a new species and a new ichnospecies from the Carboniferous of New England, USA, Zoological Journal of the Linnean Society, zlad088, https://doi.org/10.1093/zoolinnean/zlad088

https://academic.oup.com/zoolinnean/advance-article/doi/10.1093/zoolinnean/zlad088/7241373?login=true

Should be "UC: Upper Cretaceous", not "UC: Upper Cretceous"


TABLE 2

Title is incorrect, should be "Table 2 - Fossil species of Solifugae..."

For consistency "Dunlop, Erdek and Bartel" [and missing]

The mixture of Lower Cretaceous and Cennomanian is a bit odd. If the Crato Formation is Aptian/Albian you can give the stage here as, e.g., "Apt/Alb".

Your abbreviations here are also inconsistent with the ones you use in Table 1. Here you have EC (Early Cretaceous) instead of LC (Lower Cretaceous). Also the Burmese (Myanmar) amber is listed in Table 1 as (UC for Upper Cretaceous), but in Table 2 as "Cen" for Cenomanian.

In general it is not good to mix Upper/Lower and Early/Late when referring to geological ages. Please try to standardize your geological ages (Period/Stage) and their abbreviations across both tables.



I do not mind my identity being revealed to the authors.

Jason Dunlop, Berlin

Reviewer 2 ·

Basic reporting

The English needs some polishing to make it more readable and grammatically correct.

There are numerous places where contractions are used, e.g. it’s, isn’t. Contractions should not be used in scientific writing.

The illustrations are very nice. However, it would be extremely useful, and follow usual paleontological practice, to have camera lucida drawings alongside the photos to help the reader identify the various parts.

In the abstract, the information about how the specimens got to the museum, while important, is unnecessary in an abstract. An abstract should be a simple, succinct summary of the science: the problem, the methodology, and the results ONLY.

Experimental design

There needs to be a section under Mesoproctus rayoli n. sp. called Diagnosis, in which the details of how the new species differs from Mesoproctus rowlandi are listed. This is standard zoological practice in naming new organisms. At present, this information is buried in Remarks.

It would be helpful to have further discussion on the evidence for the arid environment. For example, the salt pseudomorphs (mentioned) and the abundant beds of gypsum in surrounding sediments, suggest hypersaline water and arid hinterland. See, for example: Downen, M. R., Selden, P. A. & Hasiotis, S. T. 2016. Spider leg flexure as an indicator of salinity in lacustrine paleoenvironments. Palaeogeography, Palaeoclimatology, Palaeoecology 445: 115–123.

Validity of the findings

I am delighted to see the extensive collections of Crato fossils in the Santana do Cariri Museum now being described by the scientific community.

Annotated reviews are not available for download in order to protect the identity of reviewers who chose to remain anonymous.

---

## Round 0.2 · Minor Revisions

Please, read the reviewer's suggestion regarding Diagnosis.

·

Basic reporting

Still needs a little polishing of the English to make it more readable, but that should be a simple job for the Editor.

Experimental design

No experiments, but good investigation

Validity of the findings

Everything OK

Additional comments

The authors have now included the obligatory Diagnosis. However, they should not include an explicit measurement for the body size. By stating "Body length (including pygidium) 65.9 mm", it means that a new specimen with a body length of, say, 65.5 mm could be construed as a new species because it does not conform to this diagnosis! Much better to say "Large Mesoproctus (body length including pygidium around 66 mm)"

Since the Diagnosis is generally used to distinguish the new species from its congeners (in this case there is only one), you could be more specific, for example::

"The new species differs from Mesoproctus rowlandi Dunlop, 1998 in being considerably larger (~66 mm body length including pygidium v. ~25 mm for the type species); by the pedipalp coxal apophysis having one terminal tooth, one accessory tooth on the inner margin, and outer margin of the apophysis serrated proximally (v. endites of coxa of pedipalps without ornamentation in M. rowlandi); in the proportions of the opisthosoma: distinctly oval in M. rayoli n. sp. (v. cylindrical in M. rowlandi); and leg IV is much longer than the opisthosoma in M. rayoli n. sp. (v. the leg IV just reaching or slightly overreaching the pygidium in M. rowlandi)."

Also, I want to commend the authors on producing a beautiful reconstruction of the suggested live appearance of these animals in their natural habitat!

---

## Round 0.3 · accepted · Accept

Thank you very much for submitting to PeerJ, and congratulations!